

# Speciated atmospheric mercury and sea-air exchange of
# gaseous mercury in the South China Sea
Chunjie Wang[1], Zhangwei Wang[1], Fan Hui[2], Xiaoshan Zhang[1]

[1] Research Center for Eco-Environmental Sciences, Chinese Academy of Sciences, 18 Shuangqing Road, Beijing,
China
[2] China University of Petroleum (Beijing), 18 Fuxue Road, Beijing, China

Correspondence to: Xiaoshan Zhang (zhangxsh@rcees.ac.cn)



**Abstract**


The characteristics of the reactive gaseous mercury (RGM) and particulate mercury ($Hg^P$) in the
marine boundary layer (MBL) is poorly understood due in part to sparse data from sea and ocean.
Gaseous elemental Hg (GEM), RGM and size-fractioned $Hg^P$ in marine atmosphere, and dissolved
gaseous Hg (DGM) in surface seawater were determined in the South China Sea (SCS) during an
oceanographic expedition (3−28 September 2015). The mean concentrations of GEM, RGM and
$Hg^P_{2.5}$ were $1.52 \pm 0.32$ ng m$^{-3}$, $6.1 \pm 5.8$ pg m$^{-3}$ and $3.2 \pm 1.8$ pg m$^{-3}$, respectively. Low GEM
level indicated that the SCS suffered less influence from human activities, which could be due to
the majority of air masses coming from the open oceans as modeled by backward trajectories.
Atmospheric reactive Hg (RGM + $Hg^P_{2.5}$) represented less than 1 % of total atmospheric Hg,
indicating that atmospheric Hg existed mainly as GEM in the MBL. The GEM and RGM
concentrations in the northern SCS were significantly higher than those in the western SCS, and
the $Hg^P_{2.5}$ and $Hg^P_{10}$ levels in the Pearl River Estuary were significantly higher than those in the
open waters of the SCS, indicating that the Pearl River Estuary was polluted to some extent. The
size distribution of $Hg^P$ in $PM_{10}$ was observed to be bi-modal with a higher peak (5.8−9.0 μm) and
a lower peak (0.7−1.1 μm), but the coarse modal was the dominant size, especially in the open
SCS. There was no significant diurnal variation of GEM and $Hg^P_{2.5}$, but we found the RGM
concentrations were significantly higher in daytime than in nighttime mainly due to the influence
of solar radiation. In the northern SCS, the DGM concentrations in nearshore areas were higher
than those in the open sea, but this pattern was not significant in the western SCS. The sea-air
exchange fluxes of $Hg^0$ in the SCS varied from 0.40 to 12.71 ng m$^{-2}$ h$^{-1}$ with a mean value of 4.99
$\pm 3.32$ ng m$^{-2}$ h$^{-1}$. The annual emission flux of $Hg^0$ from the SCS to the atmosphere was estimated
to be 159.6 tons yr$^{-1}$, accounting for about 5.54 % of the global $Hg^0$ oceanic evasion though the
SCS only represents 1.0 % of the global ocean area. Additionally, the annual dry deposition flux of
atmospheric reactive Hg represented more than 18 % of the annual evasion flux of $Hg^0$, and
therefore the dry deposition of atmospheric reactive Hg was an important pathway for the input of
atmospheric Hg to the SCS.
**1   Introduction**
Mercury (Hg) is a naturally occurring metal. Hg is released to the environment through both the
natural and anthropogenic pathways (Schroeder and Munthe, 1998). However, since the Industrial
Revolution, the anthropogenic emissions of Hg increased drastically. Continued rapid
industrialization has made Asia the largest source region of Hg emissions to air, with East and
Southeast Asia accounting for about 40 % of the global total (UNEP, 2013). Three operationally
defined Hg forms are present in the atmosphere: gaseous elemental Hg (GEM or $Hg^0$), reactive
gaseous Hg (RGM) and particulate Hg ($Hg^P$) (Schroeder and Munthe, 1998; Landis et al., 2002),
while they have different physicochemical characteristics. GEM is very stable with a residence



time of 0.2−1 yr due to its high volatility and low solubility (Weiss-Penzias et al., 2003; Radke et
al., 2007; Selin et al., 2007; Horowitz et al., 2017). Therefore, GEM can be transported for a
long-range distance in the atmosphere, and this makes it well-mixed on a regional and global scale.
Generally, GEM makes up more than 95 % of total atmospheric Hg (TAM), while the RGM and
$Hg^P$ concentrations (collectively known as atmospheric reactive mercury) are typically 2−3 orders
of magnitude smaller than GEM in part because they are easily removed from ambient air by wet
and dry deposition (Laurier and Mason, 2007; Holmes et al., 2009; Gustin et al., 2013), and they
can also be reduced back to $Hg^0$.

Numerous previous studies have shown that $Hg^0$ in the marine boundary layer (MBL) can be

rapidly oxidized to form RGM in situ (Hedgecock et al., 2003; Laurier et al., 2003; Sprovieri et al.,
2003, 2010; Laurier and Mason, 2007; Soerensen et al., 2010a; Wang et al., 2015). Ozone and OH
could potentially be important oxidants on aerosols (Ariya et al., 2015), while the reactive halogen
species (e.g., Br, Cl and BrO, generating from sea salt aerosols) may be the dominant sources for
the oxidation of $Hg^0$ in the MBL (Laurier et al., 2003; Sander et al., 2003; Holmes et al., 2006,
2010; Seigneur and Lohman, 2008; Auzmendi-Murua et al., 2014; Gratz et al., 2015; Steffen et al.,
2015; Shah et al., 2016; Horowitz et al., 2017). The wet and dry deposition (direct or uptake by
sea-salt aerosol) represents a major input of RGM and $Hg^P$ to the sea and ocean due to their
special and unique characteristics (i.e., high reactivity and water solubility) (Lindberg and Stratton,
1998; Landis et al., 2002; Mason and Sheu, 2002; Holmes et al., 2009). Previous studies also
showed that atmospheric wet and dry deposition of RGM (mainly $HgBr_2$, $HgCl_2$, HgO,
Hg-nitrogen and sulfur compounds) was the greatest source of Hg to open oceans (Mason and
Sheu, 2002; Holmes et al., 2009; Mason et al., 2012; Huang et al., 2017). A recent study suggested
that approximately 80 % of atmospheric reactive Hg sinks into the global oceans, and most of the
deposition takes place to the tropical oceans (Horowitz et al., 2017).

The atmospheric reactive Hg deposited to the oceans follows different reaction pathways, and

one important process is that divalent Hg can be combined with the existing particles followed by
sedimentation, or be converted to methylmercury (MeHg), the most bioaccumulative and toxic
form of Hg in seafood (Ahn et al., 2010; Mason et al., 2017), another important process is that the
divalent Hg can be converted to dissolved gaseous Hg (DGM) through abiotic and biotic
mechanisms (Fitzgerald et al., 2007; Strode et al., 2007). It is well known that almost all DGM in
the surface seawater is $Hg^0$ (Mason et al., 1995; Horvat et al., 2003), while the dimethylmercury is
extremely rare in the surface seawater (Hammerschmidt et al., 2012; Bowman et al., 2015). It has
been found that a majority of the surface seawater was supersaturated with respect to $Hg^0$
(Fitzgerald et al., 2007; Soerensen et al., 2010b, 2013, 2014), and parts of this $Hg^0$ may be emitted
to the atmosphere. Evasion of $Hg^0$ from the oceanic surface into the atmosphere is partly driven by
the solar radiation and aquatic Hg pools of natural and anthropogenic origins (Andersson, et al.,
2011). Sea-air exchange is an important component of the global Hg cycle as it mediates the rate



of increase in ocean Hg and therefore the rate of change in level of MeHg. Consequently, $Hg^0$
evasion from sea surface not only decreases the amount of Hg available for methylation in waters
but also has an important effect on the redistribution of Hg in the global environment (Mason and
Sheu, 2002; Strode et al., 2007).
In recent years, speciated atmospheric Hg has been monitored in coastal areas (e.g., Choi et
al., 2008; Cheng et al., 2013, 2014; Xu et al., 2015; Mao et al., 2016; Howard et al., 2017) and
open seas and oceans (e.g., Laurier and Mason, 2007; Chand et al., 2008; Soerensen et al., 2010a;
Wang et al., 2016a, b). However, there exists a dearth of knowledge regarding speciated
atmospheric Hg and sea-air exchange of $Hg^0$ in tropical seas, such as the South China Sea (SCS).
The highly time-resolved ambient GEM concentrations were measured using a Tekran® system.
Simultaneously, the RGM, $Hg^P$ and DGM were measured using manual methods. The main
objectives of this study are to identify the spatial-temporal characteristics of speciated atmospheric
Hg and to investigate the DGM concentrations in the SCS during the cruise, and then to calculate
the $Hg^0$ flux based on the meteorological parameters as well as the concentrations of GEM in air
and DGM in surface seawater. These results will raise our knowledge of the Hg cycle in tropical
marine atmosphere and waters.

## 2   Materials and methods

### 2.1   Study area

The SCS is located in the downwind of Southeast Asia (Fig. 1a), and it is the largest semi-enclosed
marginal sea in the western tropical Pacific Ocean. The SCS is connected with the East China Sea
(ECS) to the northeast and the western Pacific Ocean to the east (Fig. 1a). The SCS is surrounded
by numerous developing and developed countries (see Fig. 1a). An open cruise was organized by
the South China Sea Institute of Oceanology (Chinese Academy of Sciences) and conducted
during the period of 3–28 September 2015. The sampling campaign was conducted on R/V *Shiyan*
*3*, which departed from Guangzhou, circumnavigated the northern and western SCS and then
returned to Guangzhou. The DGM sampling stations and R/V tracks are plotted in Fig. 1b. In this
study, meteorological parameters (including photosynthetically available radiation (PAR)
(Li-COR®, Model: Li-250), wind speed, air temperature and RH) were measured synchronously
with atmospheric Hg onboard the R/V.

### 2.2   Experimental methods

#### 2.2.1   Atmospheric GEM measurements

In this study, GEM was measured using an automatic dual channel, single amalgamation cold
vapor atomic fluorescence analyzer (Model 2537B, Tekran®, Inc., Toronto, Canada), which has
been reported in our previous studies (Wang et al., 2016a, b, c). In order to reduce the



contamination from ship exhaust plume as possible, we installed the Tekran® system inside the
ship laboratory (the internal air temperature was controlled to 25 ℃ using an air conditioner) on
the fifth deck of the R/V and mounted the sampling inlet at the front deck 1.5 m above the top
deck (about 16 m above sea level) using a 7 m heated (maintained at 50 ℃)
polytetrafluoroethylene (PTFE) tube (¼ inch in outer diameter). The sampling interval was 5 min
and the air flow rate was 1.5 l min$^{-1}$ in this study. Moreover, two PTFE filters (0.2 μm pore size,
47 mm diameter) were positioned before and after the heated line, and the soda lime before the
instrument was changed every 3 days during the cruise. The Tekran® instrument was calibrated
every 25 h using the internal calibration source and these calibrations were checked by injections
of certain volume of saturated Hg$^0$ before and after this cruise. The relative percent difference
between manual injections and automated calibrations was < 5 %. The precision of the analyzer
was determined to > 97 %, and the detection limit was < 0.1 ng m$^{-3}$.

The meteorological and basic seawater parameters were collected onboard the R/V, which

was equipped with meteorological and oceanographic instrumentations. To investigate the
influence of air masses movements on the GEM levels, 72-h backward trajectories of air masses
were calculated using the Hybrid Single Particle Lagrangian Integrated Trajectory (HYSPLIT)
model (Draxler and Rolph, 2012) and TrajStat software (Wang et al., 2009) based on Geographic
Information System. Global Data Assimilation System (GDAS) meteorological dataset
(ftp://arlftp.arlhq.noaa.gov/pub/archives/gdas1/) with 1 ° × 1 ° latitude and longitude horizontal
spatial resolution and 23 vertical levels at 6-h intervals was used as the HYSPLIT model input. It
should be noted that the start time of each back trajectory was identical to the GEM sampling time
(UTC) and the start height was 500 m above sea level.
**2.2.2    Sampling and analysis of RGM and Hg$^P$**
The Hg$^P_{2.5}$ (Hg$^P$ in PM$_{2.5}$) was collected on quartz filter (47 mm in diameter, Whatman), which has
been reported in several previous studies (Landis et al., 2002; Liu et al., 2011; Kim et al., 2012;).
It should be pointed out that the KCl coated denuders were heated at 500 ℃ for 1 h and the quartz
filters were pre-cleaned by pyrolysis at 900 ℃ for 3 h to remove the possible pollutant. The RGM
and Hg$^P_{2.5}$ were sampled using a manual system (URG-3000M), which has been reported in
previous studies (Landis et al., 2002; Liu et al., 2011; Wang et al., 2016b). The sampling unit
includes an insulated box (Fig. S1), two quartz annular denuders, two Teflon filter holder (URG
Corporation) and a pump etc. The sampling flow rate was 10 l min$^{-1}$ (Landis et al., 2002), and the
sampling inlet was 1.2 m above the top deck of the R/V. In this study, one Hg$^P_{2.5}$ sample was
collected in the daytime (6:00−18:00) and the other in the nighttime (18:00−6:00 (next day)),
while two RGM samples were collected in the daytime (6:00−12:00 and 12:00−18:00, local time)
and one RGM sample in the nighttime. Quality assurance and quality control for Hg$^P$ and RGM
were carried out using field blank samples and duplicates. The field blank denuders and quartz





filters were treated similarly to the other samples but not sampling. The mean relative differences
of duplicated $Hg^P_{2.5}$ and RGM samples (n = 6) were 13 ±6 % and 9 ±7 %, respectively.

Meanwhile, we collected different size particles using an Andersen impactor (nine-stage),

which has been widely used in previous studies (Feddersen et al., 2012; Kim et al., 2012; Zhu et
al., 2014; Wang et al., 2016a). The Andersen cascade impactor was installed on the front top deck
of the R/V to sample the size-fractioned particles in $PM_{10}$. In order to diminish the contamination
from exhaust plume of the ship as much as possible, we turned off the pump when R/V arrived at
stations, and then switched back on when the R/V went to next station. The sample collection
began in the morning (10:00 am) and continued for 2 days with a sampling flow rate of 28.3 l
$min^{-1}$. Field blanks for $Hg^P$ were collected by placing nine pre-cleaned quartz filters (81 mm in
diameter, Whatman) in another impactor for 2 days without turning on the pump. After sampling,
the quartz filters were placed in cleaned plastic boxes (sealing in Zip Lock plastic bags), and then
were immediately preserved at −20 ℃ until the analysis.

The detailed analysis processes of RGM and $Hg^P$ have been reported in our previous studies

(Wang et al., 2016a, b). Briefly, the denuder and quartz filter were thermally desorbed at 500 ℃
and 900 ℃, respectively, and then the resulting thermally decomposed $Hg^0$ in carrier gas (zero air,
i.e., Hg-free air) was quantified. The method detection limit was calculated to be 0.67 pg $m^{-3}$ for
RGM based on 3 times the standard deviation of the blanks (n = 57) for the whole dataset. The
average field blank of denuders was 1.2 ± 0.6 pg (n = 6). The average blank values (n = 6) of
$Hg^P_{2.5}$ and $Hg^P_{10}$ were 1.4 pg (equivalent of < 0.2 pg $m^{-3}$ for a 12 h sampling time) and 3.2 pg
(equivalent of < 0.04 pg $m^{-3}$ for a 2-day sampling time) of Hg per filter, respectively. The
detection limits of $Hg^P_{2.5}$ and $Hg^P_{10}$ were all less than 1.5 pg $m^{-3}$ based on 3 times the standard
deviation of field blanks. It should be noted that the average field blanks for RGM and $Hg^P$ were
subtracted from the samples.
**2.2.3   Determination of DGM in surface seawater**
In this study, the analysis was carried out according to the trace element clean technique, all
containers (borosilicate glass bottles and PTFE tubes, joints and valves) were cleaned prior to use
with detergent, followed by trace-metal-grade $HNO_3$ and HCl, and then rinsed with Milli-Q water
(> 18.2 MΩ $cm^{-1}$), which has been described in our previous study (Wang et al., 2016c). DGM
were measured in situ using a manual method (Fu et al., 2010; Ci et al., 2011). The detailed
sampling and analysis of DGM has been elaborated in our previous study (Wang et al., 2016c).
The analytical blanks were conducted onboard the R/V by extracting Milli-Q water for DGM. The
mean concentration of DGM blank was 2.3 ± 1.2 pg $l^{-1}$ (n = 6), accounting for 3−10 % of the raw
DGM in seawater samples. The method detection limit was 3.6 pg $l^{-1}$ on the basis of three times
the standard deviation of system blanks. The relative standard deviation of duplicate samples
generally < 8 % of the mean concentration (n = 6).





### 215  2.2.4  Estimation of sea-air exchange flux of Hg$^0$

The sea-air flux of Hg$^0$ was calculated using a thin film gas exchange model developed by Liss
and Slater (1974) and Wanninkhof (1992). The detailed calculation processes of Hg$^0$ flux have
been reported in recent studies (Ci et al., 2011; Kuss, 2014; Wang et al., 2016c; Kuss et al., 2018).
It should be noted that the Schmidt number for gaseous Hg ($Sc_{Hg}$) is defined as the following
equation: $Sc_{Hg} = v/D_{Hg}$, where $v$ is the kinematic viscosity (cm$^2$ s$^{-1}$) of seawater calculated using
the method of Wanninkhof (1992), $D_{Hg}$ is the Hg$^0$ diffusion coefficient (cm$^2$ s$^{-1}$) in seawater,
which is calculated according to the recent research (Kuss, 2014). The degree of Hg$^0$ saturation ($S_a$)
was calculated using the following equation: $S_a = H' DGM_{conc}/GEM_{conc}$, and the calculation of $H'$
(the dimensionless Henry's Law constant) has been reported in previous studies (Ci et al., 2011,
2015; Kuss, 2014).

### 226  3  Results and discussion

### 227  3.1  Speciated atmospheric Hg concentrations

Figure 2 shows the time series of speciated atmospheric Hg and meteorological parameters during
the cruise in the SCS. The GEM concentration during the whole study period ranged from 0.92 to
4.12 ng m$^{-3}$ with a mean value of 1.52 $\pm$ 0.32 ng m$^{-3}$ (n = 4673), which was comparable to the
average GEM level over the global open oceans (Soerensen et al., 2010a), and higher than those at
background sites in the Southern Hemisphere  (Slemr et al., 2015; Howard et al., 2017), and also
higher than those in remote oceans, such as the Cape Verde Observatory station (Read et al., 2017),
the Atlantic Ocean (Laurier and Mason, 2007; Soerensen et al., 2013), the equatorial Pacific
Ocean (Soerensen et al., 2014) and the Indian Ocean (Witt et al., 2010; Angot et al., 2014), but
lower than those in marginal seas, such as the Bohai Sea (BS), Yellow Sea (YS) and East China
Sea (ECS) (Table 1). However, previous studies (Fu et al., 2010; Tseng et al., 2012) conducted in
the northern SCS showed that the average GEM concentrations in their study period (Fu et al.,
2010; Tseng et al., 2012) were higher than that in this study (Table 1). This is due to the fact that
the GEM level in the northern SCS (Fu et al., 2010; Tseng et al., 2012) were considerably higher
than that in the western SCS (this study).
The Hg$^P_{2.5}$ concentrations over the SCS ranged from 1.2 to 8.3 pg m$^{-3}$ with a mean value of
3.2 $\pm$ 1.8 pg m$^{-3}$ (n = 39) (Fig. 2), which was higher than those observed at Nam Co (China) and
the Amsterdam Island, and were comparable to those in other coastal areas, such as the Okinawa
Island, the Nova Scotia, the Adriatic Sea, the Ontario lake and the Weeks Bay (see Table 1), but
lower than those in the BS and YS (Wang et al., 2016b), and considerably lower than those in
rural and urban sites, such as Xiamen, Seoul (see Table 1), Guiyang and Waliguan (Fu et al., 2011,
2012). The results showed that the SCS suffered less influence from human activities. The RGM
concentration over the SCS ranged from 0.27 to 27.57 pg m$^{-3}$ with a mean value of 6.1 $\pm$ 5.8 pg



m$^{-3}$ (n = 58), which was comparable to those in other seas, such as the North Pacific Ocean, the
North Atlantic Ocean and the Mediterranean Sea (including the Adriatic Sea) (Table 1), and higher
than the global mean RGM concentration in the MBL (Soerensen et al., 2010a), and also higher
than those measured at a few rural sites (Valente et al., 2007; Liu et al., 2010; Cheng et al., 2013,
2014), but significantly much lower than those polluted urban areas in China and South Korea,
such as Guiyang (35.7 $\pm$43.9 pg m$^{-3}$, Fu et al., 2011), Xiamen, and Seoul (Table 1). Furthermore,
Figure 2 shows that the long-lived GEM has smaller variability compared to the short-lived
species like RGM and Hg$^P_{2.5}$, indicating that atmospheric reactive Hg was easily scavenged from
the marine atmosphere due to their high activity and solubility. This pattern was consistent with
our previous observed patterns in the BS and YS (Wang et al., 2016b). Moreover, we found that
atmospheric reactive Hg represents less than 1 % of TAM in the atmosphere, which was
comparable to those measured in other marginal and inner seas, such as the BS and YS (Wang et
al., 2016b), Adriatic Sea (Sprovieri and Pirrone, 2008), Okinawa Island (located in the ECS)
(Chand et al., 2008), but was significantly lower than those at the urban sites (Table 1).
**3.2 Spatial distribution of atmospheric Hg**
**3.2.1 Spatial distributions of GEM and RGM**
The spatial distribution of GEM over the SCS is illustrated in Fig. 3a. The mean GEM
concentration in the northern SCS (1.73 $\pm$ 0.40 ng m$^{-3}$ with a range of 1.01–4.12 ng m$^{-3}$) was
significantly higher than that in the western SCS (1.41 $\pm$0.26 ng m$^{-3}$ with a range of 0.92–2.83 ng
m$^{-3}$) (*t*-test, $p < 0.01$). Additionally, we found that the GEM concentrations in the PRE (the
average value > 2.00 ng m$^{-3}$) were significantly higher than those in the open SCS (see Figs. 2, 3a),
indicating that this nearshore area suffered from high GEM pollution in our study period probably
due to the surrounding human activities. Figure 3a shows that there was large difference in GEM
concentration between stations 1–10 and stations 16–31. The 72-h back-trajectories of air masses
showed that the air masses with low GEM levels between stations 1 and 10 mainly originated
from the SCS (Fig. S2a), while the air masses with high GEM levels at stations 16–31 primarily
originated from East China and ECS, and then passed over the southeast coastal regions of China
(Fig. S2b). Additionally, Fig. 3a shows that there was small variability of GEM concentrations
over the western SCS except the measurements near the station 79. The back-trajectories showed
that the air masses with elevated GEM level near the station 79 originated from the south of the
Taiwan Island, while the other air masses mainly originated from the West Pacific Ocean (Fig. S3a)
and the Andaman Sea (Fig. S3b). Therefore, the air masses dominantly originated from sea and
ocean in this study period, and this could be the main reason for the low GEM level over the SCS.
In conclusion, GEM concentrations showed a conspicuous dependence on the sources and
movement patterns of air masses during this cruise. In addition to the anthropogenic emissions, the
emission of Hg$^0$ from the surface seawater may be another important source of Hg$^0$ to the





atmosphere (Ci et al., 2011; Soerensen et al., 2013, 2014), especially for this tropical sea.
The spatial distribution of RGM over the SCS is plotted in Fig. 3b. The mean RGM
concentration in the northern SCS (7.1 $\pm$ 1.4 pg m$^{-3}$) was also obviously higher than that in the
western SCS (3.8 $\pm$ 0.7 pg m$^{-3}$) ($t$-test, $p < 0.05$), indicating that a portion of RGM in the northern
SCS maybe originated from the anthropogenic emission. We observed elevated RGM
concentrations in the PRE, and which was consistent with the GEM distribution pattern, indicating
that part of the RGM near PRE probably originated from the surrounding human activities. This is
confirmed by the following fact: The RGM concentrations in nighttime of the two days in the PRE
were 11.3 and 5.2 pg m$^{-3}$ (Fig. S3), and they were significantly higher than those in the open SCS.
Another obvious feature is that the amplitude of RGM concentration is much greater than the
GEM, and this further indicated that the RGM was easily removed from the atmosphere through
both the wet and dry deposition. In addition, we found that the RGM concentrations in the
nearshore area were not always higher than those in the open sea except the measurements in the
PRE, suggesting that the RGM in the remote marine atmosphere presumably not originated from
land but from the in situ photo-oxidation of Hg$^0$, which had been reported in previous studies (e.g.,
Hedgecock and Pirrone, 2001; Lindberg et al., 2002; Laurier et al., 2003; Sprovieri et al., 2003,
2010; Sheu and Mason, 2004; Laurier and Mason, 2007; Soerensen et al., 2010a; Wang et al.,

2015).

**3.2.2    Spatial distributions of Hg$^P_{2.5}$ and Hg$^P_{10}$**
The concentrations and spatial distribution of Hg$^P_{2.5}$ in the MBL are illustrated in Fig. 4a. The
highest Hg$^P_{2.5}$ value (8.3 pg m$^{-3}$) was observed in the PRE during daytime on 4 September 2015
presumably because of the local human activities. The homogeneous distribution and lower level
of Hg$^P_{2.5}$ in the open SCS indicated that the Hg$^P_{2.5}$ not originated from the land and the SCS
suffered less influence from human activities especially in the open sea. This is due to the fact that
the majority of air masses in the SCS during this study period came from the seas and oceans. The
spatial distribution pattern of Hg$^P_{2.5}$ in this study was different from our previous observed
patterns in the BS and YS (Wang et al., 2016b), which showed that Hg$^P_{2.5}$ concentrations in
nearshore area were higher than those in the open sea both in spring and fall mainly due to the
outflow of atmospheric Hg$^P$ from East China.
The concentrations and spatial distributions of Hg$^P_{10}$ in the MBL of the SCS are illustrated in
Fig. 4b. We found that the Hg$^P_{10}$ concentration was considerably (2−7 times) higher in the PRE
than those of other regions of the SCS probably due to the large emissions of anthropogenic Hg in
surrounding areas of the PRE. Moreover, the highest Hg$^P_{2.1}$/Hg$^P_{10}$ ratio (41 %) was observed in the
PRE and coastal sea area of Hainan Island, while lowest ratio (22 %) was observed in the open sea
(Fig. 4b). The Hg$^P_{10}$ concentrations and Hg$^P_{2.1}$/Hg$^P_{10}$ ratios were higher in the nearshore area
compared to those in the open sea, demonstrating that coastal sea areas are polluted by





anthropogenic Hg to a certain extent. Interestingly, we found the mean $Hg^P_{2.1}$ concentration (3.16
± 2.69 pg m$^{-3}$, n = 10) measured using the Andersen sampler was comparable to the mean $Hg^P_{2.5}$
concentration (3.33 ± 1.89 pg m$^{-3}$, n = 39) measured using a 47 mm Teflon filter holder (*t*-test, *p* >
0.1). This indicated that the fine $Hg^P$ level in the MBL of the SCS was indeed low, and there might
be no significant difference in $Hg^P$ concentration in the SCS between 12 h and 48 h sampling time.

The concentrations of all size-fractioned $Hg^P$ are summarized in Table S1. The size

distribution of $Hg^P$ in the MBL of the SCS is plotted in Fig. 5. One striking feature is that the
bi-modal pattern (a higher peak (5.8−9.0 μm) and a lower peak (0.7−1.1 μm)) was observed for
the size distributions of $Hg^P$ in the open sea (Fig. 5a) if we excluded the data in the PRE. The
bi-modal pattern was more obvious when we consider all the data (Fig. 5b). Generally, the $Hg^P$
concentrations in coarse particles were significantly higher than those in fine particles, and $Hg^P_{2.1}$
contributed approximately 32 % (22−41 %, see Fig. 4b) to the $Hg^P_{10}$ for the whole data, indicating
that the coarse mode was the dominant size during this study period. This might be explained by
the sources of the air masses. Since air masses dominantly originated from sea and ocean (Figs. S1,
S2) and contained high concentrations of sea salts which generally exist in the coarse mode (1−10
μm) (Athanasopoulou et al., 2008; Mamane et al., 2008), the $Hg^P_{2.1}/Hg^P_{10}$ ratios were generally
lower in the SCS compared to those in the BS, YS and ECS (Wang et al., 2016a).
**3.3   Dry deposition fluxes of RGM and $Hg^P$**
The dry deposition flux of $Hg^P_{10}$ was obtained by summing the dry deposition fluxes of each
size-fractionated $Hg^P$ in the same set. The dry deposition flux of $Hg^P_{10}$ is calculated using the
following equation: $F = \sum CHg^P \times V_d$, the $F$ is the dry deposition flux of $Hg^P_{10}$ (ng m$^{-2}$ d$^{-1}$), $CHg^P$
is the concentration of $Hg^P$ in each size fraction (pg m$^{-3}$), and $V_d$ is the corresponding dry
deposition velocity (cm s$^{-1}$). In this study, the dry deposition velocities of 0.03, 0.01, 0.06, 0.15
and 0.55 cm s$^{-1}$ (Giorgi, 1988; Pryor et al., 2000; Nho-Kim et al., 2004) were chosen for the
following size-fractioned particles: < 0.4, 0.4−1.1, 1.1−2.1, 2.1−5.8 and 5.8−10 μm, respectively
(Wang et al., 2016a). The average dry deposition flux of $Hg^P_{10}$ was estimated to be 1.08 ng m$^{-2}$ d$^{-1}$
based on the average concentrations of each size-fractionated $Hg^P$ in the SCS (Table S2), which
was lower than those in the BS, YS and ECS (Wang et al., 2016a). The dry deposition velocity of
RGM was 4.0–7.6 cm s$^{-1}$ because of its characteristics and rapid uptake by sea salt aerosols
followed by deposition (Poissant et al., 2004; Selin et al., 2007). The annual dry deposition fluxes
of $Hg^P_{10}$ and RGM to the SCS were calculated to be 1.42 and 27.39–52.05 tons yr$^{-1}$ based on the
average $Hg^P_{10}$ and RGM concentrations and the area of the SCS (3.56 × 10$^{12}$ m$^2$). The result
showed that RGM contributed more than 95 % to the total dry deposition of atmospheric reactive
Hg. The annual dry deposition flux of RGM was considerably higher than that of the $Hg^P_{10}$ due to
the higher deposition rate and concentrations of RGM.
**3.4   Temporal variation of atmospheric Hg**





### 3.4.1   diurnal variation of GEM

The diurnal variation of GEM concentration during the whole study period is illustrated in Fig. 6.
It was notable that there was no significant variability of the mean ($\pm$ SD) GEM concentration in a
whole day during this study period, and the GEM concentration dominantly fell in the range of
1.3−1.7 ng m$^{-3}$ (Fig. 6). The statistical result showed that the mean GEM concentration in the
daytime (6:00−18:00) (1.49 $\pm$ 0.06 pg m$^{-3}$) was comparable to that in the nighttime (1.51 $\pm$ 0.06
pg m$^{-3}$) (t-test, $p > 0.05$). The lower GEM concentrations and smaller variability over the SCS
further revealed that the SCS suffered less influence of human activities, and the evasion of DGM
in local or regional surface seawater of the SCS and surrounding oceans was probably an
important source for the GEM in the marine atmosphere.

### 3.4.2   Daily variation of RGM

The average RGM concentrations in the daytime and nighttime are illustrated in Fig. 7. Firstly, it
could be found that RGM showed a diurnal variation with higher concentrations in the daytime
and lower concentrations in the nighttime during the whole study period. The mean RGM
concentration in the daytime (8.0 $\pm$ 5.5 pg m$^{-3}$) was significantly and considerably higher than that
in the nighttime (2.2 $\pm$ 2.7 pg m$^{-3}$) (t-test, $p < 0.001$). This diurnal pattern was in line with the
previous multiple sites studies (Laurier and Mason, 2007; Liu et al., 2007; Engle et al., 2008;
Cheng et al., 2014). This is due to the fact that the oxidation of GEM in the MBL must be
photochemical, which have been evidenced by the diurnal cycle of RGM (Laurier and Mason,
2007). Another reason is that there was more Br (gas phase) production during daytime (Sander et
al., 2003). Figure S3 showed that the RGM concentration in the nighttime was lower than those in
corresponding forenoon and afternoon except the measurements in the PRE. This further indicated
that (1) the RGM originated from the photo-oxidation of Hg$^{0}$ in the atmosphere and (2) the RGM
was easily and quickly removed from the atmosphere in nighttime.

In addition, we found that the difference in RGM concentration between day and night in the

SCS was higher than those in the BS and YS (Wang et al., 2016b), and one possible reason is that
the solar radiation and air temperature over the SCS were stronger and higher compared to those
over the BS and YS (Wang et al., 2016b) as a result of the specific location of the SCS (tropical
sea) and the different sampling time (the SCS: September 2015, the BS and YS: April−May and
November 2014). Secondly, it could be found that the higher the RGM concentrations in the
daytime, and the higher the RGM concentrations in the nighttime, but the concentrations in
daytime were higher than that in the corresponding nighttime throughout the sampling period (see
Figs. 7, S3). This is partly because the higher RH and lower air temperature in nighttime were
conductive to the removal of RGM (Rutter and Schauer, 2007; Amos et al., 2012). Thirdly, we
found that the difference in RGM concentration between different days was large though there
was no significantly difference in PAR values (Fig. 7). However, here again divide two kinds of





cases: the first kind of circumstance is that the higher RGM in the PRE (day and night)
presumably mainly originated from the surrounding human activities (i.e., 4−5 September 2015);
the second scenario is that RGM in open waters mainly originated from the in situ oxidation of
GEM in the MBL (Soerensen et al., 2010a; Sprovieri et al., 2010). The main reason for the large
difference in RGM concentration between different days was that there was large difference in
wind speed and RH between different days (see Fig. 2), and the discussion can be found in the
following paragraphs.

### 401  3.4.3  Daily variation of $Hg^P_{2.5}$

Figure 8 shows the $Hg^P_{2.5}$ concentrations in the daytime and nighttime during the entire study
period. The $Hg^P_{2.5}$ value in the daytime ($3.4 \pm 1.9$ pg m$^{-3}$, n = 20) was slightly but not significantly
higher than that in the nighttime ($2.4 \pm 0.9$ pg m$^{-3}$, n = 19) (*t*-test, $p > 0.1$), and this pattern was
consistent with the result of our previous study conducted in the open waters of YS (Wang et al.,
2016b). The higher $Hg^P_{2.5}$ concentrations in the PRE and nearshore area of the Hainan Island (Fig.
4 and Fig. 8) indicated that the nearshore areas were readily polluted due to the anthropogenic Hg
emissions, while the lower $Hg^P_{2.5}$ level in the open sea further suggested that the open areas of the
SCS suffered less anthropogenic $Hg^P$. Therefore, we postulate that the $Hg^P_{2.5}$ over the open SCS
mainly originated from the in situ formation. During the cruise in the western SCS (16−28
September 2015), we found elevated $Hg^P_{2.5}$ concentrations when the RGM concentrations were
high at lower wind speed (e.g., 20−22 September 2015, it was sunny all these days) (see Figs. 2, 7,
8). This is probably due to the transferring of RGM from the gas to the particle phase. In contrast,
we found that the $Hg^P_{2.5}$ concentrations were elevated when the RGM concentrations were low at
higher wind speed (e.g., 25−27 September 2015, it was cloudy these days, and there was a
transitory drizzly on 26 September 2015) (see Figs. 2, 7, 8). On the one hand, high wind speed
may increase the levels of halogen atoms (Br and Cl etc.) and sea salt aerosols in the marine
atmosphere, which in turn were favorable to the production of RGM and formation of $Hg^P_{2.5}$
(Auzmendi-Murua et al., 2014); on the other hand, high wind speed was favorable to the removal
of RGM and $Hg^P_{2.5}$ in the atmosphere, this was probably the reason for lower RGM and $Hg^P_{2.5}$
concentrations during 25−27 September as compared to those observed during 20−22 September
(see Fig. 2).
Pearson's correlation coefficients were calculated between speciated Hg and meteorological
parameters to identify the relationships between them (Table 2). According to the correlation
analysis, the $Hg^P_{2.5}$ was significantly positively correlated with RGM. Part of the reason was that
RGM could be adsorbed by particulate matter under high RGM concentrations and then enhanced
the $Hg^P$ concentrations. Similarly, the $Hg^P_{2.5}$ had a significantly positive correlation with GEM, on
the one hand, GEM and $Hg^P$ probably originated from the anthropogenic sources especially in the
PRE and nearshore areas; on the other hand, it was probably due to the fact that GEM could be



oxidized to form RGM and then Hg$^P$, which might be the reason for the positive but not
significant correlation between RGM and GEM since higher GEM level may result in higher
RGM level in daytime. The correlation analysis showed that the Hg$^P_{2.5}$ and RGM were all
negatively correlated with wind speed and RH (Table 2), and the higher wind speed was favorable
to the removal of Hg$^P_{2.5}$ over the RGM. This is because the high wind speed might increase the
RH levels and then elevated wind speed and RH may accelerate the removal of Hg$^P_{2.5}$ and RGM
(Cheng et al., 2014; Wang et al., 2016b). Moreover, both the air temperature and PAR were
positively correlated with RGM and Hg$^P_{2.5}$, and a significantly positive correlation was found
between PAR and RGM, indicating that the role of solar radiation played on the production of
RGM was more obvious than that on the formation of Hg$^P_{2.5}$, which were consistent with the
previous study at coastal and marine sites (Mao et al., 2012).
**3.5  Sea-air exchange of Hg$^0$ in the SCS**
The spatial distributions of DGM and Hg$^0$ fluxes in the SCS are illustrated in Fig. 9. The DGM
level in nearshore area was higher than that in the open sea, and this pattern was similar to our
previous study conducted in the ECS (Wang et al., 2016c). The DGM concentration in this study
varied from 23.0 to 66.8 pg l$^{-1}$ with a mean value of 37.1 $\pm$ 9.0 pg l$^{-1}$ (Fig. 9a and Table S3),
which was higher than those in other open oceans, such as the Atlantic Ocean (Anderson et al.,
2011), the West Atlantic Ocean and the South Pacific Ocean (Soerensen et al., 2013, 2014), but
considerably lower than that in the Minamata Bay (Marumoto et al., 2015). The mean DGM
concentration in the northern SCS (41.3 $\pm$ 10.9 pg l$^{-1}$) was significantly higher than that in the
western SCS (33.5 $\pm$ 5.0 pg l$^{-1}$) (*t*-test, $p < 0.01$). The reason was that DGM concentrations in the
nearshore areas of the PRE and Hainan Island were higher than those in the western open sea (see
Fig. 9a). The DGM in surface seawater of the SCS was supersaturated with a saturation of 501 %
to 1468 % with a mean value of 903 $\pm$ 208 %, which was approximately two thirds of that
measured in the ECS (Wang et al., 2016c). The result indicated that (1) the surface seawater in the
SCS was supersaturated with gaseous Hg and (2) Hg$^0$ evaporated from the surface seawater to the
atmosphere during our study period.
The sea-air exchange fluxes of Hg$^0$ at each station are presented in Table S3, including GEM,
DGM, PAR, surface seawater temperature, wind speed and saturation of Hg$^0$. Sea-air exchange
fluxes of Hg$^0$ in the SCS ranged from 0.40 to 12.71 ng m$^{-2}$ h$^{-1}$ with a mean value of 4.99 $\pm$ 3.32
ng m$^{-2}$ h$^{-1}$ (Fig. 9b and Table S3), and which was comparable to the previous measurements
obtained in the Mediterranean Sea, the northern SCS and the West Atlantic Ocean (Andersson et
al., 2007; Fu et al., 2010; Soerensen et al., 2013), but lower than those in polluted marine
environments, such as the Minamata Bay, the Tokyo Bay and the YS (Narukawa et al., 2006; Ci et
al., 2011; Marumoto et al., 2015), while higher than those in some open sea environments, such as
the Baltic Sea, the Atlantic Ocean and the South Pacific Ocean (Kuss and Schneider, 2007;

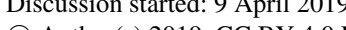

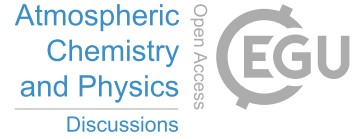

Andersson et al., 2011; Kuss et al., 2011; Soerensen et al., 2014). Interestingly, we found the $Hg^0$
flux near the station 99 were higher than those in open water as a result of higher wind speed
(Table S3). In order to better understand the important role of the SCS, we relate the $Hg^0$ flux in
the SCS to the global estimation, an annual sea-air flux of $Hg^0$ was calculated based on the
assumption that there was no seasonal variation in $Hg^0$ emission flux from the SCS. The annual
emission flux of $Hg^0$ from the SCS was estimated to be 159.6 tons $yr^{-1}$ assuming the area of the
SCS was $3.56 \times 10^{12}\, m^2$ (accounting for about 1.0 % of the global ocean area), which constituted
about 5.5 % of the global $Hg^0$ oceanic evasion (Strode et al., 2007; Soerensen et al., 2010b; UNEP,
2013). We attributed the higher $Hg^0$ flux in the SCS to the specific location of the SCS (tropical
sea) and the higher DGM concentrations in the SCS (especially in the northern area). Therefore,
the SCS may actually play an important role in the global Hg oceanic cycle. Additionally, we
found that the percentage of the annual dry deposition flux of atmospheric reactive Hg to the
annual evasion flux of $Hg^0$ was approximately 18−34 %, indicating that the dry deposition of
atmospheric reactive Hg was an important pathway for the atmospheric Hg to the ocean.
**4   Conclusions**
During the cruise aboard the R/V *Shiyan 3* in September 2015, GEM, RGM and $Hg^P$ were
determined in the MBL of the SCS. The GEM level in the SCS was comparable to the background
level over the global oceans due to the air masses dominantly originated from seas and oceans.
GEM concentrations were closely related to the sources and movement patterns of air masses
during this cruise. Moreover, the speciated atmospheric Hg level in the PRE was significantly
higher than those in the open SCS due to the anthropogenic emissions. The $Hg^P$ concentrations in
coarse particles were significantly higher than those in fine particles, and the coarse modal was the
dominant size though there were two peaks for the size distribution of $Hg^P$ in $PM_{10}$, indicating that
most of the $Hg^P_{10}$ originated from in situ production. There was no significant difference in GEM
and $Hg^P_{2.5}$ concentrations between day and night, but RGM concentrations were significantly
higher in daytime than in nighttime. RGM was positively correlated with PAR and air temperature,
but negatively correlated with wind speed and RH. The DGM concentrations in nearshore areas of
the SCS were higher than those in the open sea, and the surface seawater of the SCS was
supersaturated with respect to $Hg^0$. The annual flux of $Hg^0$ from the SCS accounted for about
5.5 % of the global $Hg^0$ oceanic evasion though the area of the SCS just represents 1.0 % of the
global ocean area, suggesting that the SCS plays an important role in the global Hg cycle.
Additionally, the dry deposition of atmospheric reactive Hg was a momentous pathway for the
atmospheric Hg to the ocean because it happens all the time.

**5   Appendix A**





**Table A1** List of acronyms and symbols

| Abbreviation | Full name |
|---|---|
| BS | Bohai Sea |
| YS | Yellow Sea |
| ECS | East China Sea |
| SCS | South China Sea |
| PRE | Pearl River Estuary |
| MBL | Marine boundary layer |
| GEM | Gaseous elemental mercury |
| RGM | Reactive gaseous mercury |
| TAM | Total atmospheric mercury |
| $Hg^P_{2.1}$ | Particulate mercury in $PM_{2.1}$ |
| $Hg^P_{2.5}$ | Particulate mercury in $PM_{2.5}$ |
| $Hg^P_{10}$ | Particulate mercury in $PM_{10}$ |
| DGM | Dissolved gaseous mercury |


Data are available from the first author Chunjie Wang (888wangchunjie888@163.com).
*Author contributions.* XZ and ZW designed the study. CW and FH organized the mercury
measurements. CW performed the data analysis, and wrote the paper. All authors contributed to
the manuscript with discussions and comments.
*Competing interests.* The authors declare that they have no conflict of interest.
*Acknowledgments.* This research was funded by the National Basic Research Program of China
(No. 2013CB430002), National Natural Science Foundation of China (No. 41176066) and
"Strategic Priority Research Program" of the Chinese Academy of Sciences, Grant No.
XDB14020205. We gratefully acknowledge the open cruise organized by the South China Sea
Institute of Oceanology, Chinese Academy of Sciences. The technical assistance of the staff of the
R/V *Shiyan 3* is gratefully acknowledged.

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

## 788    Figures and Tables

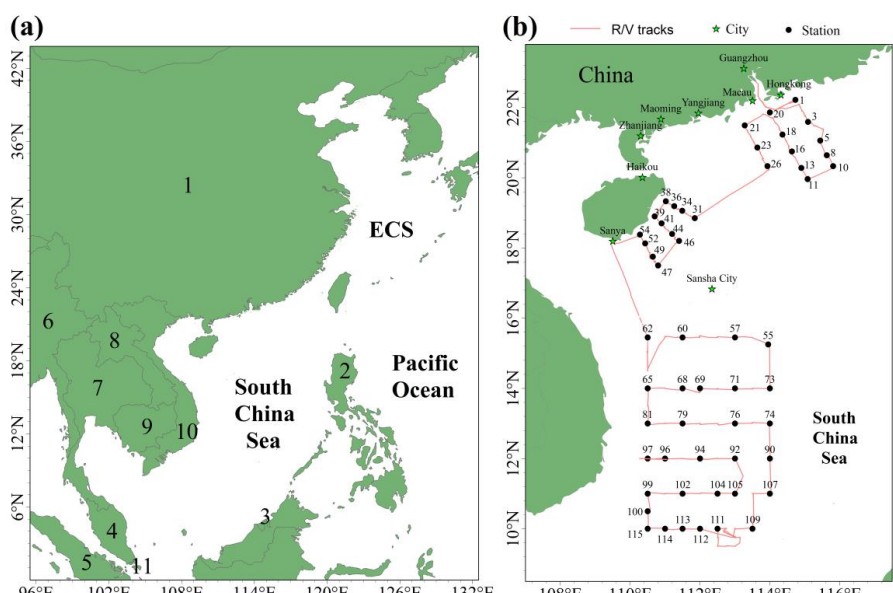

**Figure 1.** Map of the South China Sea (a) (1: China, 2: Philippines, 3: Brunei, 4: Malaysia, 5:
Indonesia, 6: Myanmar, 7: Thailand, 8: Laos, 9: Cambodia, 10: Vietnam, 11: Singapore), and
locations of the DGM sampling station and the R/V tracks (b). It should be noted that the black
solid points represent the sampling stations, and the number near the black solid point represents
the name of the station.





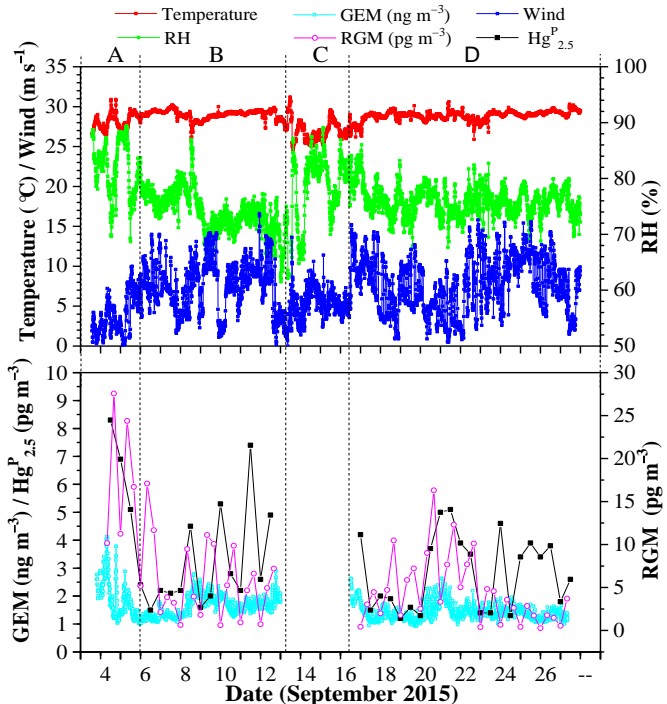


**Figure 2.** Time (local time) series of GEM, $Hg^P_{2.5}$, RGM and some meteorological parameters,
including relative humidity (RH), air temperature and wind speed ("A" represents the data
measured in the Pearl River Estuary (PRE), "B" represents the data measured in the northern SCS,
"C" represents the data obtained in the port of Sanya, "D" represents the data measured in the
western SCS). It was rainy day on the days of 8 and 26 September 2015.





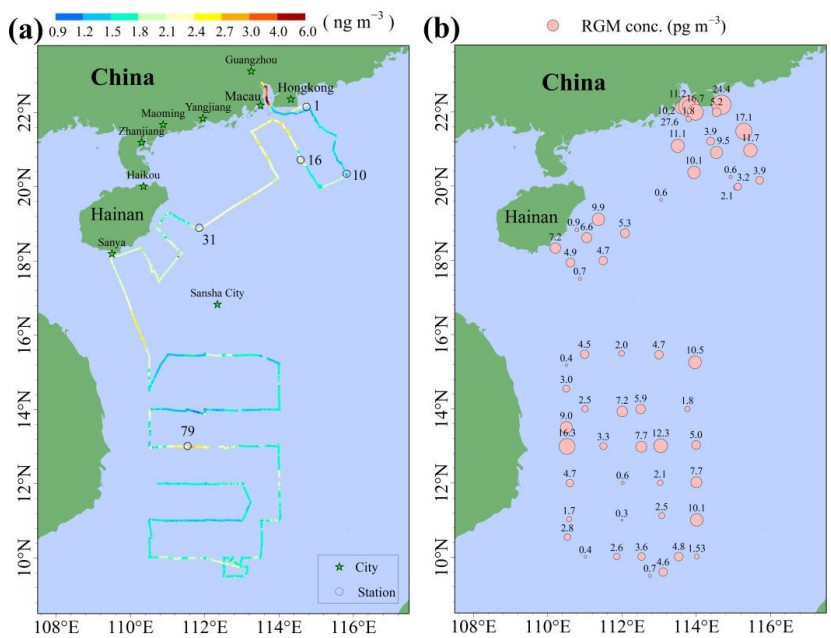

**Figure 3.** The concentrations and spatial distributions of GEM (a) and RGM (b) in the MBL of the SCS.

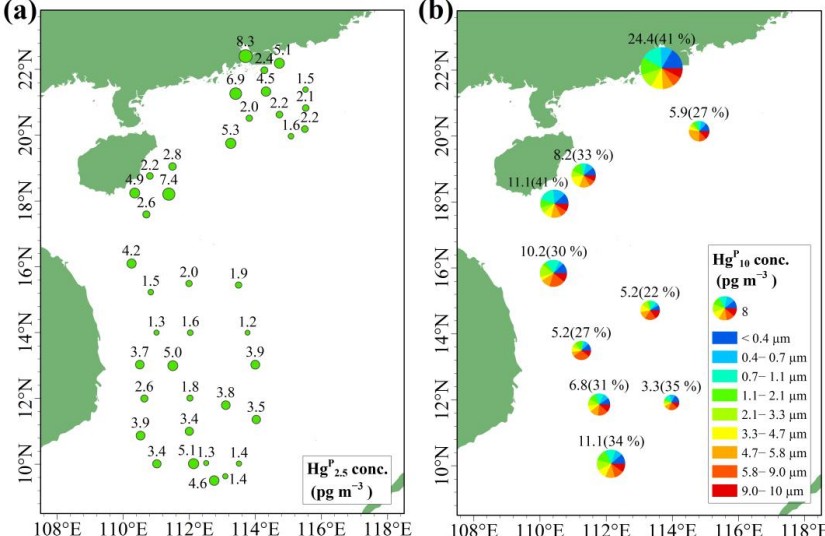

**Figure 4.** Spatial distributions of $Hg^P_{2.5}$ (a) and $Hg^P_{10}$ ($Hg^P_{2.1}$/$Hg^P_{10}$ ratio) (b) in the MBL of the SCS. $Hg^P_{2.5}$, $Hg^P_{2.1}$ and $Hg^P_{10}$ denote the $Hg^P$ in $PM_{2.5}$, $PM_{2.1}$ and $PM_{10}$, respectively.



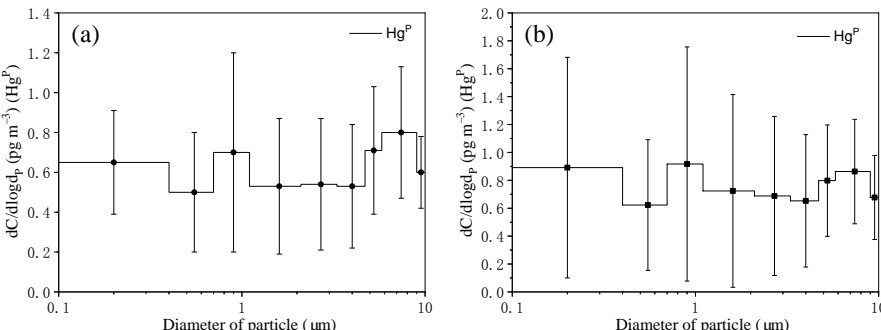


**Figure 5.** Size distributed concentrations of Hg$^P$ (PM$_{10}$) in the MBL of the SCS, (a) represents all
the data excepting the measurements in the PRE; (b) represents all the data. The data shown are
the mean and standard error.

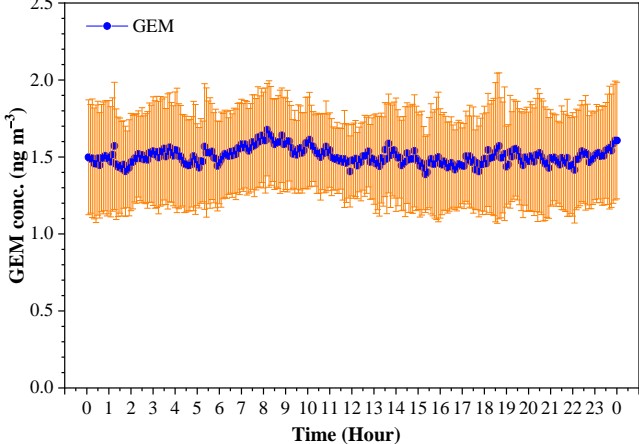


**Figure 6.** Diurnal variation of GEM concentration (mean $\pm$ SD) over the SCS.




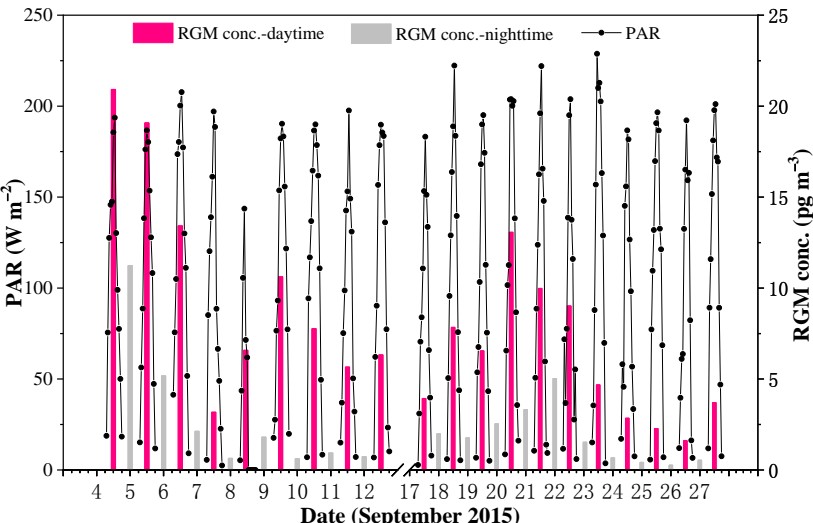


**Figure 7.** Daily variation of RGM concentration over the SCS.


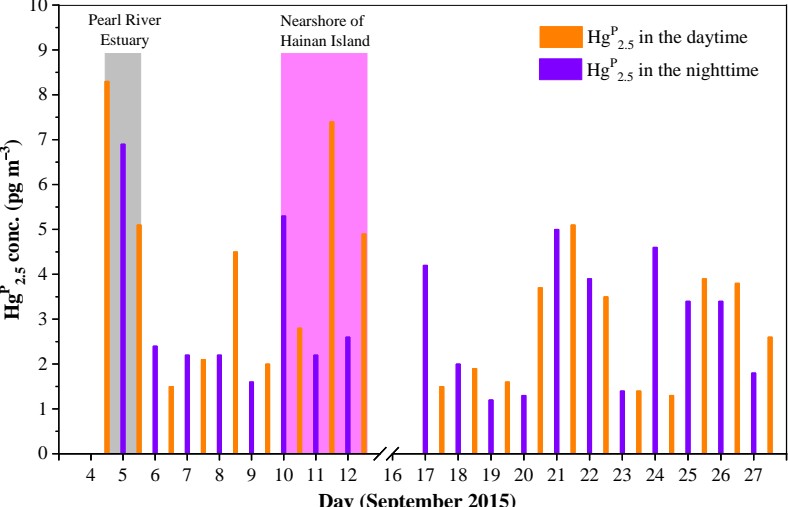


**Figure 8.** Daily variation of $Hg^P_{2.5}$ in the MBL of the SCS. The light gray area represents the data

in the PRE, while the light magenta area represents the data in the nearshore area of the Hainan

Island.





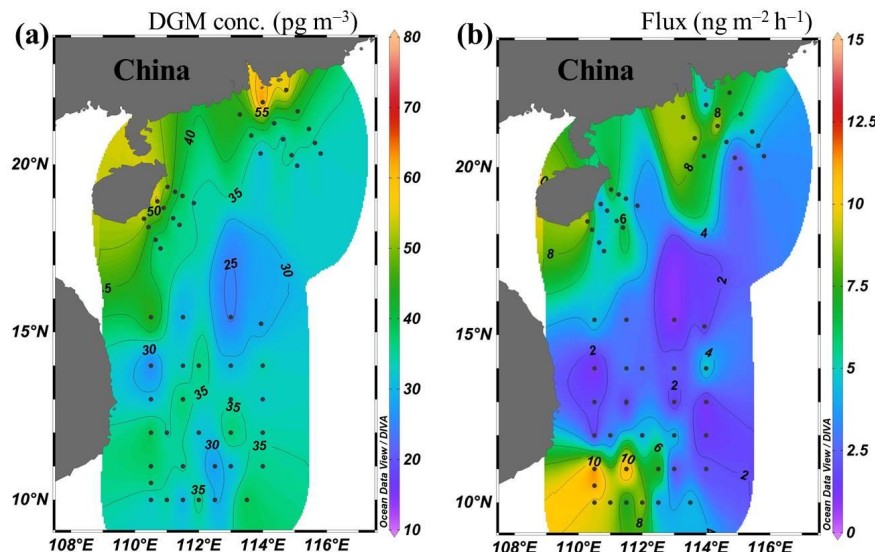

**Figure 9.** Spatial distributions of DGM (a) and sea-air exchange flux of $Hg^0$ (b) in the SCS.

**Table 1.** The GEM, $Hg^P_{2.5}$ and RGM concentrations in this study and other literature.

| Location | | Classification | Sampling time | GEM (ng m$^{-3}$) | $Hg^P_{2.5}$ (pg m$^{-3}$) | RGM (pg m$^{-3}$) | Reference |
|---|---|---|---|---|---|---|---|
| China | SCS | Sea | 2015 | 1.52 ±0.32 | 3.2 ±1.8 | 6.1 ±5.8 | This study |
| | BS and YS | Sea | 2014 (Spring) | 2.03 ±0.72 | 11.3 ±18.5 | 2.5 ±1.7 | Wang et al., 2016a, b |
| | BS and YS | Sea | 2014 (Fall) | 2.09 ±1.58 | 9.0 ±9.0 | 4.3 ±2.5 | Wang et al., 2016a, b |
| | YS | Sea | 2010 (Summer) | 2.61 ±0.50 | NA$^a$ | NA | Ci et al., 2011 |
| | YS | Sea | 2012 (Spring) | 1.86 ±0.40 | NA | NA | Ci et al., 2015 |
| | YS | Sea | 2012 (Fall) | 1.84 ±0.50 | NA | NA | Ci et al., 2015 |
| | ECS | Sea | 2013 (Summer) | 1.61 ±0.32 | NA | NA | Wang et al., 2016c |
| | ECS | Sea | 2013 (Fall) | 2.20 ±0.58 | NA | NA | Wang et al., 2016c |
| | Northern SCS | Sea | 2007 | 2.62 ±1.13 | NA | NA | Fu et al., 2010 |
| | Northern SCS | Sea | 2003−2005 | 2.8 −5.7 | NA | NA | Tseng et al., 2012 |
| | Nam Co | lake | 2014−2015 | 0.95 ±0.37 | 0.85 ±2.91 | 49.0 ±60.3 | de Foy et al., 2016 |
| | Xiamen | Coastal urban | 2012−2013 | 3.50 | 61.05 | 174.41 | Xu et al., 2015 |
| Japan | Okinawa Island | Ocean | 2004 | 2.04 ±0.38 | 3.0 ±2.5 | 4.5 ±5.4 | Chand et al., 2008 |
| Korea | Seoul | Urban | 2005−2006 | 3.22 ±2.10 | 23.9 ±19.6 | 27.2 ±19.3 | Kim et al., 2009 |
| USA | Weeks Bay | Coast | 2005−2006 | 1.6 ±0.3 | 2.7 ±3.4 | 4.0 ±7.5 | Engle et al., 2008 |
| Canada | Ontario Lake | Remote area | 2005−2006 | 1.57 ±0.22 | 4.42 ±3.67 | 0.99 ±1.89 | Cheng et al., 2012 |
| | Nova Scotia | Coast | 2010−2011 | 1.67 ±1.01 | 2.32 ±3.09 | 2.07 ±3.35 | Cheng et al., 2013 |
| | Nova Scotia | Coast-rural | 2010−2011 | 1.38 ±0.20 | 3.5 ±4.5 | 0.4 ±1.0 | Cheng et al., 2014 |
| Australia | ATARS$^b$ | Coast | 2014-2015 | 0.95 ±0.12 | NA | NA | Howard et al., 2017 |
| South-west India Ocean | | Ocean | 2007 | 1.24 ±0.06 | NA | NA | Witt et al., 2010 |
| North Atlantic Ocean | | Ocean | 2003 | 1.63 ±0.08 | NA | 5.9 ±4.9 | Laurier et al., 2007 |
| West Atlantic Ocean | | Ocean | 2008−2010 | 1.4−1.5 | NA | NA | Soerensen et al., 2013 |
| North Pacific Ocean | | Ocean | 2002 | 2.5 | NA | 9.5 | Laurier et al., 2003 |
| Pacific Ocean | | Ocean | 2011 | 1.15−1.32 | NA | NA | Soerensen et al., 2014 |
| Mediterranean Sea | | Sea | 2000 | 1.9 ±1.0 | NA | 7.9 | Sprovieri et al., 2003 |
| Global Ocean | | Ocean | 2006−2007 | 1.53 ±0.58 | NA | 3.1 ±11.0 | Soerensen et al., 2010a |
| Adriatic Sea | | Ocean | 2004 | 1.6 ±0.4 | 4.5 ±8.0 | 6.7 ±11.7 | Sprovieri and Pirrone, 2008 |
| Amsterdam Island | | Ocean | 2012−2013 | 1.03 ±0.08 | 0.67 | 0.34 | Angot et al., 2014 |

$^a$ NA: No data available.

$^b$ ATARS: Australian Tropical Atmospheric Research Station.



**Table 2.** Correlation coefficients for speciated atmospheric Hg and meteorological parameters (one asterisk

denotes significant correlation in $p < 0.05$, double asterisks denotes significant correlation in $p < 0.01$).

| Speciation | GEM | | RGM | | $Hg^P_{2.5}$ | | Wind speed | | Air temperature | | RH | | PAR | |
|---|---|---|---|---|---|---|---|---|---|---|---|---|---|---|
| | $p$ | $r$ | $p$ | $r$ | $p$ | $r$ | $p$ | $r$ | $p$ | $r$ | $p$ | $r$ | $p$ | $r$ |
| RGM | 0.069 | 0.294 | | | < 0.01 | 0.453** | 0.123 | −0.251 | 0.053 | 0.313 | 0.065 | −0.299 | < 0.01 | 0.638** |
| $Hg^P_{2.5}$ | < 0.01 | 0.539** | < 0.01 | 0.489** | | | 0.037 | −0.335* | 0.621 | 0.082 | 0.434 | −0.129 | 0.432 | 0.130 |

833