# Peer review of "Speciated atmospheric mercury and sea-air exchange of"

_Atmospheric Chemistry and Physics, 2019_

## Referee Comment (RC1) · Anonymous Referee #1 · 4 May 2019

The authors presented a set of valuable data and conducted a meaningful analysis of the data. I have a few comments, which may help improve clarity in some places. I don't view that it is reviewers' responsibility to copy-edit and hence I did not point out all grammatical errors, but the manuscript needs careful editing.

Some of the results need to be quantitative. For instance, in the abstract, how much higher were GEM and RGM concentrations in the northern SCS, Hgp2.5 and Hgp10 in PRE than other areas (lines 48 – 50)? How much higher were RGM concentrations during the day than at night (lines 54 – 56)? How much higher were their GEM concentrations than "those background sites in the southern hemisphere" (lines 232-233)

[Figure]

and "remote oceans" (lines 234-235)? How much higher were the GEM concentrations over the northern SCS from a previous studies (lines 238-240)? They need to be quantitative about such comparisons.

Lines 81 – 88: Ye et al. (2016, acp) would be a good reference to cite, because their box model included the most up-to-date gas-phase reactions of Hg and Br and simulated contributions from variation oxidation reactions to GEM oxidation.

Line 97 – 102: Grammatical errors. They might want to break this rambling passage to three sentences.

Lines 103 – 108: Too many excess articles. In fact, this was fairly commonly throughout the text. They might want to give it a good editing to get rid of those excess articles.

Line 115: Mao et al. (2016, acp) provided a fairly complete review of the literature, up to early 2016, on spatiotemporal distributions of GEM, GOM, and PBM in different environments worldwide, including coastal areas. Not just these four studies for reference.

Lines 258-259: The larger variabilities in RGM and Hgp were due not only to scavenging but also likely due to their sensitivity to meteorological conditions and chemical environments.

Figure 3a: I suggest that the lines be thickened to make it clearer.

Please indicate where PRE is on the map. Every reader does not necessarily know where PRE is.

Lines 276, 278, 281, 282: I suspect the supplemental figure numbers were wrong. Shouldn't they be Figures S1 and S2?

Lines 330: I don't see bi-modal here. There was a third peak below 0.4 $\mu$m.

Lines 367-368: This statement needs support of evidence. I don't see where this came from.

Line 429: The GEM-Hgp correlation may also indicate the two had oceanic sources in addition to anthropogenic sources.

---

## Referee Comment (RC2) · Anonymous Referee #2 · 10 May 2019

The manuscript presents ship-based measurements of atmospheric mercury species and dissolved gaseous mercury in the Pearl River Estuary and the South China Sea. The authors used the measurements to infer the sources and sinks of elemental and reactive mercury in the atmosphere, to estimate the sea to air flux of mercury, and to assess how the mercury concentrations in the South China Sea differ from other areas.

The manuscript is well-written, and presents the results in clearly with appropriate tables and figures. The authors provide a detailed description of their measurement methods, analyze their observations systematically, and provide support for their main conclusions.

[Figure]

I do not have any major concerns about the manuscript, but a few minor comments as follows: Line 43: Here and elsewhere, I would reword "suffered less influence of human activities" to something like "less influence of fresh emissions." Line 164: The back trajectories were initiated at 500m - much higher than the measurement altitude. This needs justification. Line 172: Fig. S1 does not show the sampling unit. Line 200: How were non-detects in the RGM and HgP measurements treated? Line 331: The bimodal distribution seems less obvious in Fig. 5b. Line 351-353: I am not convinced that these 1-month observations can be extrapolated to an annual dry deposition flux. I recommend removing that calculation unless there is other evidence supporting its validity. Line 365-367: It is not obvious why the small variability in the Hg0 concentrations implies that the evasion of DGM was an important source of Hg0. Line 381: It is not clear why are higher RH and lower temperature conducive to Hg2 removal? By gas-particle partitioning?

Please mention in the main text that the acronyms are defined in the appendix.

Fig. 5: Using the same scales for the y-axes on panels a and b will be helpful.

Fig. 6: The PAR values can be removed. They do not add much information, but clutter the figure.

Fig. 9: How are the point measurements interpolated for the entire region? Was this interpolation necessary to calculate the sea-air flux? If not may be show the measurements like those in Fig. 3b.

Table 2: The correlation coefficients for HgP(2.5)-RGM and RGM-HgP(2.5) differ. Is that correct?

---

## Author Response (AR1)

**Response to Reviewer #1's Comments**

**Anonymous Referee #1:**

The authors presented a set of valuable data and conducted a meaningful analysis of the data. I have a few comments, which may help improve clarity in some places. I don't view that it is reviewers' responsibility to copy-edit and hence I did not point out all grammatical errors, but the manuscript needs careful editing.

We sincerely appreciate the reviewer's valuable comments and helpful suggestions on this manuscript. We have carefully checked the grammar, syntax and semantic of all languages throughout the manuscript based on the reviewer's suggestions. We have responded to all the comments point-by-point and made corresponding changes in the revised manuscript as highlighted in red color. Please check the detailed responses to all the comments as below. The reviewer's comments are in black and our replies are in blue.

**(1)** Some of the results need to be quantitative. For instance, in the abstract, how much higher were GEM and RGM concentrations in the northern SCS, Hgp2.5 and Hgp10 in PRE than other areas (lines 48 -50)? How much higher were RGM concentrations during the day than at night (lines 54 -56)? How much higher were their GEM concentrations than "those background sites in the southern hemisphere" (lines 232-233) and "remote oceans" (lines 234-235)? How much higher were the GEM concentrations over the northern SCS from a previous studies (lines 238-240)? They need to be quantitative about such comparisons.

Response:

We agree with the reviewer that the results should be quantitative. The concrete data has been added in the revised manuscript.

See the revised manuscript at lines 19-29, 210-218, 295, 422-423, 427-428.

**(2)** Lines 81 − 88: Ye et al. (2016, acp) would be a good reference to cite, because their box model included the most up-to-date gas-phase reactions of Hg and Br and simulated contributions from variation oxidation reactions to GEM oxidation.

Response:

Thanks for the reviewer's suggestions. We have made a careful study on the paper of "Investigation of processes controlling summertime gaseous elemental mercury oxidation at midlatitudinal marine, coastal, and inland sites". The reference has been added in the revised manuscript (lines 57-59, 62-65, 91, 750-752).

Reference:

Ye, Z., Mao, H., Lin, C.-J., and Kim, S. Y.: Investigation of processes controlling summertime gaseous elemental mercury oxidation at midlatitudinal marine, coastal, and inland sites, Atmos. Chem. Phys., 16, 8461−8478, https://doi.org/10.5194/acp-16-8461-2016, 2016.

**(3)** Line 97 − 102: Grammatical errors. They might want to break this rambling passage to three sentences.

Response:

Thanks for your suggestion. This sentence has been divided into three sentences, which has been revised as "The atmospheric reactive Hg deposited to the oceans follows different reaction pathways. One important process is that divalent Hg can be combined with the existing particles followed by sedimentation, or be converted to methylmercury (MeHg), the most bioaccumulative and toxic form of Hg in seafood (Ahn et al., 2010; Mason et al., 2017). Another important process is that the divalent Hg can be converted to dissolved gaseous Hg (DGM) through abiotic and biotic mechanisms (Strode et al., 2007)." in the revised manuscript.

Moreover, a section heading (3.5 Relationship between atmospheric Hg and meteorological parameters) has been added in the revised manuscript to make the structure of the manuscript clearer.

See the revised manuscript at lines 74-79, 401, 421.

**(4)** Lines 103 – 108: Too many excess articles. In fact, this was fairly commonly throughout the text. They might want to give it a good editing to get rid of those excess articles.
Response:
Thanks for the suggestion. We have checked all the references throughout the manuscript, and deleted those old and weakly related articles.

**(5)** Line 115: Mao et al. (2016, acp) provided a fairly complete review of the literature, up to early 2016, on spatiotemporal distributions of GEM, GOM, and PBM in different environments worldwide, including coastal areas. Not just these four studies for reference.
Response:
Thanks for your comments. We have read carefully the paper of "Current understanding of the driving mechanisms for spatiotemporal variations of atmospheric speciated mercury: a review". The related references (Ye et al., 2016 and Mao et al., 2016, 2017) have been added in the revised manuscript.
See the revised manuscript at lines 91-92.
References:
1) Ye, Z., Mao, H., Lin, C.-J., and Kim, S. Y.: Investigation of processes controlling summertime gaseous elemental mercury oxidation at midlatitudinal marine, coastal, and inland sites, Atmos. Chem. Phys., 16, 8461−8478, https://doi.org/10.5194/acp-16-8461-2016, 2016.
2) Mao, H., Cheng, I., and Zhang, L.: Current understanding of the driving mechanisms for spatiotemporal variations of atmospheric speciated mercury: a review, Atmos. Chem. Phys., 16, 12897−12924, https://doi.org/10.5194/acp-16-12897-2016, 2016.
3) Mao, H., Hall, D., Ye, Z., Zhou, Y., Felton, D., and Zhang, L.: Impacts of large-scale circulation on urban ambient concentrations of gaseous elemental mercury in New York, USA, Atmos. Chem. Phys., 17, 11655−11671, https://doi.org/10.5194/acp-17-11655-2017, 2017.

**(6)** Lines 258-259: The larger variabilities in RGM and Hgp were due not only to scavenging but also likely due to their sensitivity to meteorological conditions and chemical environments.
Response:
Thanks for the insightful comments and we do agree with the reviewer's comments. Thus, this sentence has been revised as "indicating that atmospheric reactive Hg was easily scavenged from the marine atmosphere due not only to their characteristics (high activity and solubility) but also due to their sensitivity to meteorological conditions and chemical environments" in the revised manuscript.

See the revised manuscript at lines 237-239.

**(7)** Figure 3a: I suggest that the lines be thickened to make it clearer. Please indicate where PRE is on the map. Every reader does not necessarily know where PRE is.

Response: Thanks for your suggestions. The lines have been thickened in the Figure 3a (see the revised manuscript at line 770) and Figures S2 and S3 (see the revised supplement at lines 60, 62). The location of the Pearl River Estuary (PRE) has been marked in Figures 1 and 3a (see lines 758, 770).

Moreover, the vertical heading of Figure S4 should be "RGM conc. (pg m$^{-3}$)" rather than "Hg$^P_{2.5}$ conc. (pg m$^{-3}$)", and we have corrected it (see at line 64).

**(8)** Lines 276, 278, 281, 282: I suspect the supplemental figure numbers were wrong. Shouldn't they be Figures S1 and S2?

Response:

Thanks for the reviewer's carefully check on these sentences and the Figures S1 and S2. We feel very sorry that we forgot to put Figure S1 (the picture of insulated box) in the supplement. Figure S1 has been added in the revised supplement (see at lines 58-59) of this paper. Therefore, the supplemental figure numbers were wrong in the original supplement, while the supplemental figure numbers were right in the original manuscript. We have made some modifications to ensure that the figure numbers in revised manuscript were consistent with those in revised supplement.

See the revised manuscript at lines 150, 256 258, 261-262, 273 and revised supplement at lines 58, 60, 62, 64.

**(9)** Lines 330: I don't see bimodal here. There was a third peak below 0.4 μm.

Response:

Thanks for the reviewer's carefully check on the Fig. 5. We fully agree with your comments, and we have corrected the statements in the revised manuscript.

See the revised manuscript at lines 23-24, 307-310, 468.

**(10)** Lines 367-368: This statement needs support of evidence. I don't see where this came from.

Response:

We do agree with your comment that this inference lacks sufficient evidence. Therefore, this sentence (and the evasion of DGM in local or regional surface seawater of the SCS and surrounding oceans was probably an important source for the GEM in the marine atmosphere.) has been deleted after careful consideration

**(11)** Line 429: The GEM-Hgp correlation may also indicate the two had oceanic sources in addition to anthropogenic sources.

Response:

Thanks for the in-depth comment. The sentence has been revised as "On the one hand, GEM and Hg$^P$ probably originated from the same sources (including but not limited to anthropogenic and oceanic sources) especially in the PRE and nearshore areas." in the revised manuscript.

See the revised manuscript at lines 406-408.

**Response to Reviewer #2's Comments**

**Anonymous Referee #2:**

The manuscript presents ship-based measurements of atmospheric mercury species and dissolved gaseous mercury in the Pearl River Estuary and the South China Sea. The authors used the measurements to infer the sources and sinks of elemental and reactive mercury in the atmosphere, to estimate the sea to air flux of mercury, and to assess how the mercury concentrations in the South China Sea differ from other areas. The manuscript is well-written, and presents the results in clearly with appropriate tables and figures. The authors provide a detailed description of their measurement methods, analyze their observations systematically, and provide support for their main conclusions.

I do not have any major concerns about the manuscript, but a few minor comments as follows:

We are grateful for the precise, valuable and positive comments. We have made corresponding changes in the revised manuscript as highlighted in red color. Please see the responses to the specific comments below. The reviewer's comments are in black and our replies are in blue.

**(1)** Line 43: Here and elsewhere, I would reword "suffered less influence of human activities" to something like "less influence of fresh emissions."

Response:

Thanks for your suggestion. We have changed the term "human activities" as "fresh emissions" throughout the manuscript.

See the revised manuscript at lines 15, 344.

**(2)** Line 164: The back trajectories were initiated at 500m much higher than the measurement altitude. This needs justification.

Response:

Thanks for your comment. The arrival heights of back trajectories were set at 500 m to represent the approximate height of the mixing marine boundary layer where atmospheric pollutants were well mixed. We have added "the start height was set at 500 m above sea level to represent the approximate height of the mixing marine boundary layer where atmospheric pollutants were well mixed" in the revised manuscript.

See the revised manuscript at lines 141-142.

**(3)** Line 172: Fig. S1 does not show the sampling unit.

Response:

Thanks for your carefully check on this sentence and Figure S1. The picture of the sampling unit (Figure S1) has been added in the revised supplement of this paper.

See the revised supplement at line 58.

**(4)** Line 200: How were non-detects in the RGM and HgP measurements treated?

Response:

Thanks for your comment. This sentence has been revised as "It should be noted that all the observed RGM and $Hg^P$ values were higher than the corresponding blank values, and the average blank values for RGM and $Hg^P$ were subtracted from the samples."

See the revised manuscript at lines 179-181.

**(5)** Line 331: The bimodal distribution seems less obvious in Fig. 5b.

Response:

We agree with the reviewer that the bimodal distribution seems less obvious in Fig. 5b. As a matter of fact, there were three peaks (< 0.4 μm, 0.7-1.1 μm, 5.8-9.0 μm) although the three-modal distribution was not distinct in Fig. 5. We have corrected the statements in the revised manuscript.

See the revised manuscript at lines 23-24, 307-310, 468.

**(6)** Line 351-353: I am not convinced that these 1-month observations can be extrapolated to an annual dry deposition flux. I recommend removing that calculation unless there is other evidence supporting its validity.

Response:

Thanks for your comment. Since the South China Sea (SCS) is a tropical sea (4 °N-21 °N), so we can assume that there is no large variation in the average PAR and air temperature etc. over the SCS during the four seasons. Moreover, since our sampling time spans September 2015, so the average RGM and $Hg^P$ values obtained in this sampling period can be roughly considered as the annual mean values. Additionally, Fu et al. (2010) had used the observations (August 2010) in the northern SCS to estimate the annual emission flux of $Hg^0$ over the SCS. Therefore, we think we can use the data obtained in this study to roughly estimate the dry deposition fluxes of RGM and $Hg^P$ in the SCS.

**(10)** Fig. 5: Using the same scales for the y-axes on panels a and b will be helpful.

Response:

Thanks for the suggestion. We have set the same scales for the Y-axes on panels a and b in Fig.5.

See the revised manuscript at line 776.

**(11)** Fig. 6: The PAR values can be removed. They do not add much information, but clutter the figure.

Response:

Thanks for your suggestion. There are no PAR values in Fig. 6, and I suspect that the PAR values you mentioned are the one in Fig. 7. First of all, the PAR data was essential for the following discussions. Moreover, we have changed the filled color for the RGM concentrations in nighttime. Thus, we feel the revised Fig. 7 is clearer now.

See the revised manuscript at line 782.

**(12)** Fig. 9: How are the point measurements interpolated for the entire region? Was this interpolation necessary to calculate the sea-air flux? If not may be show the measurements like those in Fig. 3b.

Response:

Thanks for your comments and suggestions. The Fig. 9 was plotted by the software of Ocean Data View based on the data obtained at each the sampling station. It is not necessary for the interpolation to calculate the sea-air flux of $Hg^0$. The DGM concentrations in surface seawater and emission fluxes of $Hg^0$ at all stations have been symbolized in the revised Fig. 9 like those in Fig. 3b.

See the revised manuscript at line 789.

**(13)** Table 2: The correlation coefficients for HgP(2.5)-RGM and RGM-HgP(2.5) differ. Is that correct?

Response:

We feel very sorry for our carelessness. After recalculation, the correlation coefficients for $Hg^P_{2.5}$-RGM and RGM-$Hg^P_{2.5}$ were same ($p < 0.01$, $r = 0.453$). We have corrected the p and r values for $Hg^P_{2.5}$-RGM in Table 2.

See the revised manuscript at lines 794-795.

[revised manuscript text omitted]